# Immune Checkpoint Inhibitors in Advanced Cutaneous Squamous Cell Carcinoma: Real-World Experience from a Canadian Comprehensive Cancer Centre

**DOI:** 10.3390/cancers15174312

**Published:** 2023-08-29

**Authors:** Erica C. Koch Hein, Maysa Vilbert, Ian Hirsch, Mauricio Fernando Ribeiro, Thiago P. Muniz, Cynthia Fournier, Khaled Abdulalem, Erick F. Saldanha, Erika Martinez, Anna Spreafico, David H. Hogg, Marcus O. Butler, Samuel D. Saibil

**Affiliations:** 1Department of Medical Oncology and Hematology, Princess Margaret Cancer Centre, University Health Network, Toronto, ON M5G 2M9, Canada; maysavilbert@gmail.com (M.V.); ian.hirsch@uhn.ca (I.H.); mauricio.ribeiro@uhn.ca (M.F.R.); thiago.muniz@uhn.ca (T.P.M.); khaled.abdulalem@uhn.ca (K.A.); erick.figueiredosaldanha@uhn.ca (E.F.S.); erika.martinez@uhn.ca (E.M.); anna.spreafico@uhn.ca (A.S.); david.hogg@uhn.ca (D.H.H.); marcus.butler@uhn.ca (M.O.B.); 2Department of Medicine, Division of Medical Oncology, University of Toronto, Toronto, ON M5S 1A8, Canada; 3Department of Hematology and Oncology, School of Medicine, Pontificia Universidad Católica de Chile, Santiago 8331150, Chile; 4Department of Medicine, Division of Dermatology, University of Toronto, Toronto, ON M5S 1A8, Canada; cynthia.fournier.2@ulaval.ca; 5Dermatology Service, Hôtel-Dieu-de-Lévis, Lévis, QC G6V 3Z1, Canada

**Keywords:** cemiplimab, pembrolizumab, immune checkpoint inhibitor, PD-1, cutaneous squamous cell carcinoma

## Abstract

**Simple Summary:**

The effectiveness and safety of immune checkpoint inhibitors in the treatment of patients with advanced cutaneous squamous cell carcinoma were evaluated in a real-world population, including patients who would have typically been excluded from clinical trials. We identified 36 patients with advanced cutaneous cell carcinoma treated with immune checkpoint inhibitors between 2017 and 2022 at a single Cancer Center in Canada; ten patients had hematological malignancy, two patients had autoimmune disease on immune suppressive drug, two patients were solid organ transplant recipients, and four patients had poor performance status. The results showed that immune checkpoint inhibitors, specifically cemiplimab and pembrolizumab, were effective and safe for advanced cutaneous squamous cell carcinoma patients, regardless of age, immune status, or performance status. Immune checkpoint inhibitors exhibit a high response rate, prolonged duration of response, and a favorable toxicity profile; these characteristics position them as a preferred therapeutic option, particularly suitable for patients with comorbidities that might otherwise preclude the utilization of conventional systemic treatments.

**Abstract:**

Immune checkpoint inhibitors (ICI) cemiplimab and pembrolizumab have revolutionized the treatment of advanced cutaneous squamous cell carcinoma (cSCC). We aimed to evaluate the effectiveness and safety of ICI in a real-world cSCC population, including patients with conditions that would exclude clinical trial participation. In this single-center, retrospective cohort study, we included all non-trial patients with advanced cSCC treated with ICI between 2017 and 2022. We evaluated investigator-assessed best overall response (BOR) and immune-related adverse events (irAEs). We correlated survival outcomes with age, performance status, immune status and irAEs. Of the 36 patients identified, the best overall response (BOR) to ICI was a partial response (PR) in 41.7%, a complete response (CR) in 27.8%, and stable disease in (SD) 13.9%. The progression-free survival (PFS) rate for 1 year was 58.1%; the median PFS was 21.3 months (95% CI 6.4–NE). The 1-year overall survival (OS) was 76.7%, and the median OS was 38.6 months (95% CI 25.4–NE). Immune-compromised patients, ECOG performance 2–3, and age ≥ 75 years were not significantly associated with PFS or OS. IrAE grades 3–4 were seen in 13.9% of patients. In our Canadian experience with real-world patients, ICI was an effective and safe treatment for advanced cSCC patients. Patients achieved great benefits with ICI regardless of age, immune status or ECOG performance status. We acknowledge the small sample size and retrospective methodology as the main limitations of our study.

## 1. Introduction

Cutaneous squamous cell carcinoma (cSCC) is one of the most common forms of skin cancer worldwide. The exact incidence rate cannot be ascertained, as most national cancer registries do not record this diagnosis [1]. cSCC traditionally accounted for the second highest incidence among skin cancer, although a recent study estimating the incidence of cSCC and basal cell carcinoma in the United States based on the total approved skin cancer treatment procedures in a population covered by Medicare cited a 1:1 ratio [2]. While the majority of cSCC are early stage and can be cured with local treatments, 3–5% of patients develop advanced disease that is not amenable to surgery or radiotherapy [3,4,5]. Moreover, as the average lifespan increases, so does the chance of encountering patients with unresectable or metastatic cSCC. A link between immune response and non-melanoma skin cancer development is supported by data showing that immunosuppression is an important risk factor for these malignancies, as well as a more aggressive course among the affected patients [6,7]. Indeed, cSCC incidence increases by tens to hundreds of times in individuals with T-cell dysfunction, such as solid organ transplant recipients, HIV-positive patients, patients with chronic lymphocytic leukemia, or other hematologic malignancies [8,9,10,11,12,13,14,15].

Programmed cell death protein 1 (PD-1) is a key immune checkpoint receptor expressed by activated T cells, and it mediates immunosuppression [16]. Inhibition of the interaction between PD-1 and its ligands PD-L1 and PD-L2 enhances T cell responses and mediates antitumor activity [17]. In the pre-immune checkpoint inhibitor (ICI) era, patients with advanced cSCC had poor long-term outcomes to palliative chemotherapy and epidermal growth factor receptor (EGFR) inhibitors [18,19,20,21,22,23]. The ICIs targeting PD-1 have revolutionized the treatment of advanced cSCC, a highly immunogenic tumor featuring a high mutational burden likely resultant from UV radiation-induced DNA damage [4,24].

The PD-1 inhibitors cemiplimab and pembrolizumab demonstrated an objective response rate of 35 to 58% and have not reached the median duration of response in phase I/II clinical trials [25,26,27,28,29,30,31]. The most extensive data on the efficacy of cemiplimab came from a phase I expansion cohort and a phase II trial of patients with unresectable locally advanced or distant metastatic cSCC [25]. In the phase I expansion cohorts, a response to cemiplimab was seen in 13 of 26 patients (50%). In the metastatic-disease cohort of the phase II study, objective responses were observed in 28 of 59 patients (47%), and 82% of these patients continued to have a response to cemiplimab at the time of data cutoff with a median follow-up of 7.9 months [25]. Cemiplimab was well tolerated, with no single grade ≥ 3 toxicity present in more than 5% of patients, and with only three patients (5%) discontinuing treatment because of toxicity. The total incidence of grade ≥ 3 treatment-related toxicity with cemiplimab was 17% [28]. In a further follow-up of a separate cohort of the phase II study mentioned above, 34 of 78 patients (44%) with locally advanced cSCC without nodal disease or distant metastasis demonstrated objective responses to cemiplimab, including 10 complete responses (13%) [26]. For this cohort, the median progression-free survival (PFS) and median overall survival (OS) had not been reached at the data cutoff. The phase II trial KEYNOTE-629 assessed the efficacy of pembrolizumab in 54 patients with locally advanced unresectable cSCC, and 105 patients with locally advanced recurrent or metastatic cSCC [29,30]. Among the patients with locally advanced unresectable disease, the objective response rate was 50% with a 17% complete response (CR) rate, and 37% of responders experienced durable responses lasting 12 months or longer. In those patients with locally advanced recurrent or metastatic disease, the objective response rate was 35% with a 10% CR rate, and 68% of responders experienced durable responses lasting 12 months or longer [29,30]. A separate phase II study, CARSKIN, assessed the efficacy of pembrolizumab in 57 patients with locally advanced unresectable or metastatic cSCC. The overall response rate was 42%, with four patients experiencing CR (7%). Pembrolizumab was well tolerated in both phase II trials, with no single grade ≥ 3 toxicity present in more than 5% of patients and to total incidence of grade ≥ 3 treatment-related toxicity of 5.7–11% [29,31]. Based on these findings, the U.S. Food and Drug Administration (FDA) approved cemiplimab in September 2018, and pembrolizumab in June 2020 for advanced cSCC.

More recently, two phase II trials assessed the efficacy of nivolumab in patients with locally advanced or metastatic cSCC [32,33]. In the first one, 14 out of 24 patients (58%) achieved an objective response, which were all partial responses; the median duration of response was not reached [32]. In the second trial, 19 out of 31 patients (61.3%) experienced an objective response, with 7 patients (22.6%) presenting a CR [33]. There were no new safety concerns. The reported incidence of grade ≥ 3 treatment-related adverse events in these two trials was 19–25% [32,33], with only one patient discontinuing nivolumab due to toxicities in the first trial [32] and two in the second [33].

The most frequent treatment related to grade 3 or 4 adverse reactions with ICI in patients with cSCC are fatigue, arthralgia, diarrhea, hyperthyroidism, rash, adrenal insufficiency, myalgia, pneumonitis and increase in liver transaminases [25,27,29,31,32,33].

Due to the recent FDA approval of these ICI therapies, limited data is available about the efficacy and safety of real-world cohorts of patients with advanced cSCC. This is especially true related to patients with poor ECOG performance status or with chronic immune suppression (e.g., use of immunosuppressant drugs for active autoimmune diseases, solid organ transplant recipients, or concomitant hematological malignancies), who were excluded from the clinical trials of ICI in cSCC [25,27,29,31,32]. The one exception was that one of the phase II trials with nivolumab allowed patients with an ECOG performance status of 2, and patients with chronic lymphocytic leukemia (CLL) that were stable under active therapy [33]. Despite this, there still is a paucity of published data describing the efficacy and toxicity of ICI in patients with cSCC and these common comorbidities. Accordingly, in this study, we investigated the clinical outcomes of patients with advanced cSCC treated with anti-PD-1 ICI outside clinical trials at a Canadian Comprehensive Cancer Centre. We aimed to assess efficacy and safety according to age, performance status, comorbidities of interest, and immune-related adverse events (irAE).

## 2. Materials and Methods

### 2.1. Study Design and Patient Cohort

In this single-center retrospective cohort study, we included all non-trial patients with incurable locoregionally advanced (defined as technically unresectable or not clinically suitable for surgery, or not amenable to radiation therapy with curative intent based on multidisciplinary tumor board discussions) or metastatic cSCC (defined as patients with disease beyond regional nodal involvement) treated with at least one dose of anti-PD-1 between June 2017 and July 2022 at a Comprehensive Cancer Centre in Canada. To identify this cohort, the institutional electronic pharmacy record was queried by diagnosis cSCC, and the name of ICI received (cemiplimab, pembrolizumab or nivolumab). Demographics and clinicopathologic features were collected from the electronic medical records in an Excel-protected dataset. Patients were staged following the 8th edition of the American Joint Committee on Cancer Staging System for cSCC of the head and neck. Treatment history, outcomes, and comorbidities of special interest (immunosuppressive condition [autoimmune disease on immunosuppressant drug, HIV, solid organ transplant recipient, hematological malignancy], and genetic syndrome predisposing to cSCC [e.g., epidermolysis bullosa]) were recorded. The data cut-off for analysis was 15 February 2023. This project was carried out with the approval of the University Health Network Research Ethics Board.

### 2.2. Efficacy and Safety Outcomes

The best overall response (BOR) was determined by the investigator´s assessment of clinical and radiological parameters and was defined as the best response recorded from the start of the treatment until disease progression. Complete response (CR) was defined as complete regression of the lesion(s); partial response (PR) as a clinical/radiological tumor reduction with persistence of detectable tumor; progression of disease (PD) as clinical/radiological increase in lesion(s) or the appearance of a new lesion; stable disease (SD) as neither response nor PD. The response rate was defined as CR + PR. The disease control rate was defined as CR + PR + SD. The investigator´s assessment of the response was performed in our facility without central review. Adverse events were assessed using Common Terminology Criteria in Solid Tumors (CTCAE) version 5.0.

Overall survival (OS) and PFS were defined, respectively, as the times from the first anti-PD-1 dose to death from any cause, and the time until the first documentation of disease progression or death from any cause, whichever occurred first. OS and PFS were censored at the date of the last follow-up.

Reason for discontinuation of treatment was also recorded: maximum benefit achieved (if treatment was stopped earlier than 2 years without any toxicities justifying treatment discontinuation and no evidence of progressive disease, at the medical oncologist´s discretion), maximum number of doses (2 years of treatment; 35 doses for cemiplimab or pembrolizumab), discontinuation due to toxicities, progression of disease or death.

For the purposes of answering our research question, we grouped patients with a disease compromising the immune system with patients with immunosuppression needs (those on immunosuppressant drugs), to whom we will refer as immunocompromised.

### 2.3. Statistical Analysis

Demographic characteristics of our cohort of patients were analyzed by descriptive statistics, such as the number of cases and percentages for discrete variables, and mean ± standard deviation or median (range) for continuous variables. Chi-square, Fisher’s exact, and Mann–Whitney U tests were used to assess differences in categorical and continuous variables among subgroups of interest, respectively. OS and PFS were estimated using the Kaplan–Meier method and the log-rank test and expressed as a median with a 95% confidence interval (CI). The univariable Cox proportional hazards model was fitted to evaluate the impact of clinical variables on survival. Multivariable analysis was not performed due to the low number of events in our cohort. Median follow-up was estimated using the Kaplan–Meier reverse method. All statistical tests were two-sided, and *p*-value < 0.05 was deemed significant. We performed all the statistical analysis in RStudio (Version 2023.03.0 + 386)

## 3. Results

### 3.1. Patients

Our cohort included 35 patients treated with cemiplimab and one with pembrolizumab; both drugs were administered every 3 weeks, as per standard of care. We did not identify any patients treated with nivolumab. Baseline patient characteristics are reported in Table 1.

At the time of ICI initiation, the median age was 75.4 years (range from 27.9 to 100.1), 36.1% of the cohort was 80 years of age or older, and 27.8% had an ECOG performance status equal to two or higher. The majority of patients were male (75%) and had another skin cancer (58%). The most common primary site of disease was the head and neck (68.6%). Sixteen of 36 (44.4%) patients had a comorbidity of interest: two with an autoimmune condition on immunosuppressant treatment, two were solid organ transplant recipients (both kidney transplant recipients), ten had hematological malignancies and two epidermolysis bullosa. None of our patients had immunosuppression related to HIV or a known HIV-positive history.

Primary treatment included surgery alone for 58.3% of patients, surgery followed by adjuvant radiation therapy for 25%, radiation therapy alone for 8.3%, and systemic therapy for 8.3%. Of those, two patients received treatment with anti-PD-1, and one with cetuximab, an EGFR inhibitor. At ICI start, most patients had an unresectable locally advanced disease (72.2%), and 27.8% had distant metastasis. Nearly all patients had a recurrent disease (91.7%), with 8.3% of patients presenting with de novo advanced disease.

All patients received single-agent anti-PD-1 in the first-line setting for advanced or metastatic disease, except for one patient who received cemiplimab after progressing after two cycles of cetuximab. Eight patients underwent radiation therapy immediately before or concurrent to ICI, either at the beginning of treatment or for treatment of oligoprogression. The median treatment duration was 10.2 months, for a median of 12 infusions (interquartile range [IQR] of 6 to 26.5), and up to a maximum of 38 cycles. Fourteen patients received ICI beyond 12 months. At the data cut-off, 10 patients (27.7%) were receiving ongoing treatment. Following anti-PD-1 discontinuation due to the progression of disease or intolerance, only two patients received a further line of systemic treatment (cetuximab). The reason for treatment discontinuation is shown in Table 2.

### 3.2. Effectiveness Outcomes

Investigator-assessed BOR of the complete cohort of patients was CR in 10 patients (27.8%), PR in 15 (41.7%), and SD in 5 (13.9%), with a disease control rate (DCR) of 83.4%. Six patients (16.7%) presented PD as the best overall response. The median treatment duration was 14.85 months for complete responders, 22.57 months for partial responders, 5.47 months for patients achieving SD as BOR, and 2.4 months for patients who presented PD as BOR (Figure 1a). When comparing the clinical activity of anti-PD1 therapy amongst various subgroups of patients, the response rate did not statistically differ according to age (age ≥ 75 versus <74), ECOG status (ECOG 0–1 versus ECOG ≥ 2) or immune status. For immune status, we compared the rate of response in patients immunocompromised (characterized by the presence of autoimmune disease on immune suppressive drugs, solid organ transplant recipient or hematological malignancy) versus immunocompetent patients that did not have these comorbidities. Interestingly, contrary to expectation, we observed significant clinical activity of the ICI therapy in patients with some immunosuppressive comorbidity. For instance, of the ten patients with hematological malignancy included in our study, four attained a CR and six achieved a PR. Similarly, for the two patients with an autoimmune disease on immune suppressive medication, one of them had a CR and the other a PR. The first patient was a 92-year-old lady with rheumatoid arthritis on methotrexate who discontinued ICI after achieving a CR. The second patient with an autoimmune disease was an 84-year-old lady with giant cell arteritis and polymyalgia rheumatica being treated with low-dose prednisone (less than 15 mg daily). She has an ongoing PR and continues on ICI treatment. Regarding our solid organ transplant recipients, one of them had a PR, and the other one had PD as BOR. The patient with PR developed aggressive pancreatic cancer and passed away. The patient with PD passed away as a consequence of cSCC progression. Overall, we observed impressive clinical activity of ICI therapy irrespective of age, ECOG or immune status. We identified two patients with epidermolysis bullosa treated with cemiplimab, and both of them achieved SD as BOR.

Additionally, many of the responses observed proved to be durable. When comparing responders (CR + PR) versus non-responders (SD + PD), the median duration of treatment was 20.2 months (95% CI 11.83–non-evaluable [NE]) versus 4.23 months (95% CI 3.43–NE) respectively (Figure 1b). Among the 10 complete responders, however, four patients did ultimately progress during follow-up. Eight patients were treated beyond the first evidence of disease progression, with none later achieving a further response.

With a median follow-up of 21.9 months (95% CI 16.4 to 23.5) among the entire cohort, 6-month PFS was 72.2% (95% CI 59–88) and 1-year PFS was 58.1% (95% CI 44–77) (Figure 2a).

The median PFS was 21.3 months (95% CI 6.4–NE). Of 15 patients who progressed, 66.7% had disease progression in the first 6 months, with only two patients progressing after 1 year. Two patients died at 6 months without disease progression. We found that patients who presented a grade 1 or 2 irAE had a higher PFS with a hazard ratio (HR) of 0.357 (95% CI 0.136–0.938, *p* = 0.028) (Figure 2b), but this did not have an impact on OS (*p* = 0.7). Conversely, patients with grade 3 or more irAE had a worse PFS and OS, although this was only statistically significant for PFS (*p* = 0.0026) (Figure 2c). We observed that patients who underwent concomitant radiation therapy had a worse PFS, with a median PFS of 4.25 months versus NE (hazard ratio [HR] 4.2, 95% CI 1.58 to 11.26, *p* = 0.0018) (Figure 2d). The PFS for patients with epidermolysis who discontinued treatment due to toxicities and disease progression were 16.4 and 4.8 months, respectively. The univariable analysis for PFS is depicted in Table 3.

The median OS was 38.6 months (95% CI 25.4–NE) among the entire cohort (Figure 3a). Twelve patients died during the follow-up period, of whom eight (66.7%) passed away in the first year. The 1-year OS was 76.7% (95% CI 0.64–0.92). For those patients who responded to ICI (CR + PR), median OS was significantly improved, as compared with non-responders (SD + PD) (38.6 versus 7.8 months, HR: 0.08, 95% CI 0.016–0.375, *p* = 0.00145) (Figure 3b). Patients with metastatic disease had a median OS of 17.1 months versus 38.6 months for patients with locally advanced disease (HR: 3.4, 95% CI 1.1 to 13.9, *p* = 0.0358 (Figure 3c).

PFS and OS were not significantly associated with age groups ≥ 75 years versus <75 years, ECOG performance status 0–1 versus 2–3, or patients with or without comorbidities of interest. The presence of distant metastatic disease increased the risk of death (HR: 3.9, 95% CI 1.1 to 13.9, *p* = 0.036). The univariable analysis for OS is depicted in Table 4.

### 3.3. Safety

PD-1 inhibition was overall well tolerated among the entire cohort, with only five patients (13.9%) developing a grade 3 or higher immune-related adverse event (irAE), which included grade 3 rash (*n* = 2), grade 4 lipase increase (*n* = 1), grade 3 fatigue (*n* = 1) and grade 3 diarrhea (*n* = 1). Toxicities led to treatment discontinuation in five patients (19.2%), and were as follows: in the first patient, grade 2 fatigue and grade 1 persistent peripheral neuropathy; in the second patient, grade 3 rash; in the third patient, grade 2 pneumonitis, in the fourth patient, grade 2 hepatitis and colitis and in the fifth patient, grade 2 polymyalgia rheumatica. Two patients died while still receiving ICI; in both cases, death was considered not related to ICI, and explained by progression of disease. At the time of data cut-off, 10 patients were still on active treatment. The main reason for ICI discontinuation was disease progression for eight patients (30.8%) (Table 2).

Out of ten patients with hematological malignancy, only one of them presented an irAE grade 3 or higher (grade 4 increase in lipase) that resolved, and the patient continues on active treatment with ICI. The two solid organ transplant recipients (kidney transplant) did not present any safety concerns, and neither of them presented allograft loss. Regarding the patients with autoimmune disease, only one of them presented a flare of their disease (polymyalgia rheumatica); this was treated with an increase in their baseline dose of prednisone, and at the moment of data cut-off their ICI treatment was ongoing. There were no safety concerns in our two patients with epidermolysis bullosa, with none of them presenting with grade 3 or more irAE. Grade 1–2 irAE (*p* = 0.1) and grade 3 or more irAE (*p =* 0.35) were similar between immunocompromised and immune-competent patients.

## 4. Discussion

Multi-disciplinary discussion of patients with advanced cSCC in tumor boards to define the oncological treatment plan is the standard approach in many oncological centers. Patients with advanced cSCC who are deemed not suitable for surgery or radiation therapy with curative intent should undergo systemic treatment with anti-PD-1 as the new standard of care. Our study confirms the efficacy and safety of anti-PD-1 inhibition in patients with advanced cSCC reported in phase I and II clinical trials [25,27,28,29,30,31,32] in a real-world setting. In contrast to previous phase I and II clinical trials [25,30,31,32], we observed a higher ORR (69.5% versus 35–58%), despite our cohort consisting of patients with both locoregional recurrent disease and distant metastasis, as opposed to some trials that also included patients with locally advanced disease only. We were surprised to observe a CR rate of 27.8% among our cohort, which is also higher than that reported in the aforementioned trials (0–16.7%). There might be several factors contributing to these differences. First, our study had a small sample size; second, the subjective nature of the investigator-assessed response and third, nearly all our patients (97.2%) had not received any previous systemic cancer treatment. Other real-world studies have reported ORR between 42% and 76.7%, and CR rates ranging from 20% to 33% [4,34,35,36,37,38,39,40,41], which is closer to our outcomes. A recent meta-analysis that included a total of 13 studies (seven randomized clinical trials and six real-world studies) with 930 patients, reported a pooled objective response rate of 47.2% [42]. We acknowledge the subjectiveness of investigator-assessed response as a main limitation of our study, which could be responsible for an overestimated response rate.

It is also important to notice that 50% of our patients would have been excluded from clinical trials; ten patients for hematological malignancy, two patients for autoimmune disease on immune suppressive drug, two patients for being recipients of solid organ transplant, and four patients for ECOG performance status ≥ 2. Our two patients with epidermolysis bullosa would also probably have been excluded from trials. Immunocompromised patients represent a uniquely challenging cohort within the population of cSCC patients. It is appreciated that immune suppression is an adverse prognostic factor in developing cSCC, with a more aggressive disease course observed in the affected individuals [6,7]. However, neither response rate nor survival outcomes differed according to immune status (compromised versus competent). Other real-world studies have described similar findings. Haist et al. described similar response rates (48.1% versus 50%, respectively) without significantly increased toxicities among immunocompromised versus immune-competent patients with advanced cSCC treated with ICI, although the remissions were often short-lived [34]. Hober et al. reported a similar response rate (50% versus 51% respectively), PFS and OS, between immunocompromised and immune competent patients [35], as well as did Hanna et al. for OS (1-year OS 56.1% for immunosuppressed versus 41.6% for immune-competent, *p* = 0.21) [39]. Taken together, we believe that ICI therapy may offer a promising treatment approach for immunocompromised advanced cSCC patients. An ongoing study is being conducted to explore the safety of PD-1 inhibition in patients with auto-immune disease and advanced metastatic or unresectable cancer (NCT03816345), and another clinical trial is studying the efficacy of combined ICIs with Tacrolimus in kidney transplant recipients with advanced melanoma and non-melanoma skin cancers (NCT03816332). The reporting from these ongoing trials will hopefully provide further clarity as to how to optimally use ICI therapy in conjunction with immune-suppressive therapies for these challenging patient populations.

Patients with epidermolysis bullosa face a significantly heightened risk of recurrent and aggressive cSCC; malignancy is often the cause of death [43]. Owing to the inherent challenges of treating epidermolysis bullosa with local therapies, there is a need to generate evidence concerning systemic treatments for this unique patient cohort. Within this population, the absence of a standardized optimal therapeutic approach for advanced cSCC is underscored by the transient nature of responses elicited by conventional chemotherapy, compounded by the absence of controlled clinical trials [44,45]. Case reports of patients with epidermolysis bullosa and cSCC have shown therapeutic efficacy and safety of anti-PD-1 agents [45,46,47], and so was our experience with two patients achieving an SD as BOR.

The median age of our cohort was 75.4 years old, which is similar to the aforementioned phase I–II clinical trials and real-world studies. The response rate and survival outcomes were similar between patients ≥ 75 years old versus younger. Even though immunosenescence may reduce the capacity of elderly patients to mediate antitumor responses [48], several studies have also shown similar response rates to ICIs among those 65 years old or older [39,49,50]. A high tumor mutational burden, associated with increased immunogenicity, is commonly observed in the tumors of older patients [51], which might be a factor contributing to the favorable response to ICIs.

Eight patients in our cohort received concomitant radiation therapy in different settings: (1) concurrent at ICI start (*n* = 3), (2) completed in the 2 weeks prior to ICI initiation (*n* = 1), or (3) concurrent for oligoprogression of disease (*n* = 4). The evidence supporting the strategy of combining radiation therapy with immunotherapy is growing [52,53]. Radiation therapy may act as an “accelerant” by killing tumor cells and triggering a systemic immune response [54]. Exposure to radiation therapy has been shown to upregulate major histocompatibility complex expression in tumor cells, prompt the recruitment of immune effector cells, and elicit systemic tumor-specific immune responses [55,56]. We observed, however, that the patients who underwent concomitant radiation therapy had a worse PFS. This might be biased and reflect the fact that in our center, patients with a more locally aggressive disease at presentation are usually offered RT concomitant to cemiplimab at the beginning of their treatment. The use of concomitant RT to ICI in these eight patients from our cohort was safe and this combination warrants further investigation. A retrospective study assessed the efficacy of pembrolizumab concurrent with RT in four patients with advanced unresectable cSCC; two patients presented a CR and 2 PD, with a median PFS of 14.4 months [57]. Two ongoing trials are exploring the efficacy of ICI in combination with RT in patients with locally advanced cSCC; one with cemiplimab (NCT05574101) and the other one with avelumab (NCT03737721).

In terms of safety, ICIs were well tolerated in our patient cohort, with only five patients (13.9%) discontinuing therapy because of toxicity. Four of the five patients who discontinued ICI due to toxicities, were older than 75 years old, with only one of them presenting a grade 3 toxicity (rash). Probably all these patients could have been rechallenged with ICI, but given their age, the occurrence of toxicities is usually less tolerated. One of the two patients with epidermolysis bullosa discontinued ICI due to toxicities (grade 1 arthralgia, peripheral sensorial neuropathy, and pruritus, and grade 2 mucositis and fatigue). Roughly 14% of the patients developed grade 3 or higher irAE, which is consistent with what has been previously reported in trials (5.7–13.9%) [26,28,29,30,31]. There were no adverse events resulting in death.

Regarding the two patients with autoimmune disease, one of them presented a flare of their disease (polymyalgia rheumatica), which was treated with an increase in their baseline dose of prednisone. As the spectrum of patients with autoimmune diseases is wide and considering that these patients are usually excluded from clinical trials, it is difficult to generate conclusions on their risk of flare-up of their autoimmune condition. Data from retrospective studies suggest that between 20 and 40% of patients experienced exacerbation of their autoimmune disease with ICI [58,59,60,61,62], which were manageable. A metanalysis of 12 studies including 193 patients with inflammatory bowel disease (IBD) treated with ICI, reported a 40% relapse of their IBD, with a third of them requiring biologic therapy [63]. As for patients with pre-existing autoimmune rheumatologic disease, patients with rheumatoid arthritis have a higher risk of flares, of about 45%, compared to those with other rheumatologic disorders [64,65]. Our safety data is similar to findings of phase I and II clinical trials with cemiplimab, pembrolizumab and nivolumab, which report grade 3 or higher treatment-related adverse events between 7 and 25% [25,27,29,31,32,33], with patients discontinuing ICI due to toxicity between 4 and 7% [25,27,31,32,33].

Despite the infrequent incidence of grade 3 or higher irAE, it is of critical significance given that advanced cSCC primarily occurs in geriatric or immunocompromised patients, where the benefits and risks of any systemic therapy necessitate careful individualized consideration. We would also like to point out that despite our aging cohort, with 47.2% of patients being older than 70 years old and 13.9% more than 90, ICIs were well tolerated. This is particularly important to note, as other forms of systemic therapy like chemotherapy may lead to increased toxicity in the elderly population. These safety results should be interpreted with caution given the retrospective nature of this study. It is important to note that the recording of adverse events in the medical chart may not have been exhaustive, which could have an impact on the accuracy of the findings. Multiple retrospective and prospective studies [66,67,68,69,70,71,72,73,74] have suggested an association between irAE and the effectiveness of ICIs, in terms of response and survival outcomes. In our cohort, we found an improved PFS in patients who presented with an irAE grade 1 or 2, but not for those who presented an irAE grade 3 or more. For patients with grade 1 or 2 irAE, there was a trend toward better OS, but it did not meet statistical significance.

Since its approval by the FDA in 2018 and 2020 for cemiplimab and pembrolizumab, respectively, both anti-PD-1 inhibitors have become preferred systemic treatment options for patients with unresectable, recurrent or metastatic cSCC according to the National Comprehensive Cancer Network Guidelines (version 1.2023). Surgery remains the cornerstone of treatment as the primary curative option for resectable patients. The therapeutic activity of neoadjuvant cemiplimab was demonstrated in a phase II non-randomized trial, which showed a 51% pathological complete response [75]. More recently, the results of the MATISSE trial were presented at the American Society of Clinical Oncology annual meeting in June 2023, which demonstrated deep responses with two infusions of neoadjuvant nivolumab alone (50%) or in combination with ipilimumab (61%) [76]. These results are encouraging, although a longer follow-up is needed to demonstrate if this finding translates into a longer disease-free survival, as it has in patients with melanoma and non-small cell lung cancer [77,78,79]. The next step will be to elucidate the role of anti-PD-1 in the adjuvant setting; two phase III clinical trials assessing adjuvant cemiplimab (NCT03969004) and adjuvant pembrolizumab (NCT03833167) following surgery and RT in locally advanced cSCC patients are currently ongoing. Finally, how to overcome primary and secondary resistance to ICI are active areas of research for many different cancer types. There are not many systemic options for patients with cSCC who progress to ICI. A phase II nonrandomized trial aiming to revert the resistance to pembrolizumab in patients with locally advanced or metastatic cSCC patients that presented SD or PD, administered cetuximab in addition to pembrolizumab until progression [80]. This study reported a response rate of 44% with the combination strategy, although grade 3–4 treatment-related adverse events occurred in 35% of patients [80]. The two patients in this study that presented acquired resistance to pembrolizumab, presented PR when introducing cetuximab [80]. In our cohort, two patients received cetuximab after progression to cemiplimab. One of them had primary resistance to cemiplimab (receiving only three cycles) and had a PR to cetuximab. The second patient had PR to cemiplimab (receiving 12 cycles) and presented a CR to cetuximab. Both patients, however, progressed to cetuximab in less than a year. A therapeutic strategy to increase the response rate of ICI and to overcome mechanisms of resistance to progression is the addition of an anti-EGFR agent. The hypothesis that the adjunct of an anti-EGFR agent could reverse the primary and secondary resistance to ICI in advanced cSCC is currently being assessed in two phase II trials: avelumab in combination with cetuximab (NCT03944941) and pembrolizumab in combination with cetuximab (NCT03666325). There are several ongoing trials exploring mechanisms to overcome resistance and improve outcomes to ICI in patients with advanced cSCC: nivolumab in combination with talimogene lapherparepvec (NCT02978625), nivolumab in combination with relatlimab (NCT04204837), pembrolizumab in combination with the C5a Antibody IFX-1 (NCT04812535), atezolizumab (anti-PD-L1) in combination with NT-I7 (recombinant IL-7-hybrid Fc, which acts through IL-7 receptor to promote proliferation, maintenance, and functionality of T-cell subsets), and intratumoral cavrotolimod (a toll-like receptor 9 agonist that activates plasmacytoid dendritic cells and triggers interferon alpha release) injections in combination with intravenous pembrolizumab or cemiplimab (NCT03684785). The results of these studies will hopefully inform further treatment approaches for patients with advanced cSCC refractory to monotherapy with an anti-PD1 ICI.

We acknowledge several limitations of our study. The small sample size, lack of comparator arm and retrospective methodology of our study may limit its ability to detect significant outcomes. Another limitation is the investigator-assessed response without a validation process, which undermines its reproducibility. Nevertheless, we also recognize the lack of clinical data on older patients or with immune suppressive conditions, which are precisely the patients with the diagnosis of advanced cSCC that we encounter in real-world scenarios. We hope to provide additional data about PD-1 inhibition from a real-world cohort of advanced cSCC patients to help clinicians in decision-making, especially for patients not represented in clinical trials.

## 5. Conclusions

The treatment of locally advanced metastatic cSCC remains a challenge, and discussion at multidisciplinary tumor boards is strongly advised. Chemotherapy and targeted agents against EGFR may achieve response in about one-third of the patients; however, these responses are not sustained, and the adverse events are often not compatible with the frailty of these patients [18,19,20,21,22,23]. ICI exhibits a high response rate, prolonged duration of response, and a favorable toxicity profile; these characteristics position them as a preferred therapeutic option, particularly suitable for elderly patients with comorbidities that might otherwise preclude the utilization of conventional systemic treatments. Our single-center institutional experience with anti-PD-1 in locally advanced or metastatic cSCC patients demonstrated its effectiveness and safety in the real-world setting, regardless of age or ECOG performance status. More important, our data suggest that patients with immunosuppressive conditions, such as active autoimmune diseases, hematological malignancies and solid organ transplant recipients, also benefit from this treatment. This is the first study to report outcomes of ICI in patients with advanced cSCC in Canada, and our findings support its use as first-line treatment. Nevertheless, new questions arise from this approach. Firstly, the quest for predictive biomarkers of response to ICI in patients with cSCC. Secondly, the optimal duration of treatment for patients attaining a complete response to ICI. Lastly, there is a need to investigate combinations that can surmount resistance mechanisms and enhance outcomes of ICI therapy; studies are ongoing to address these crucial unmet clinical needs.

## Figures and Tables

**Figure 1 cancers-15-04312-f001:**
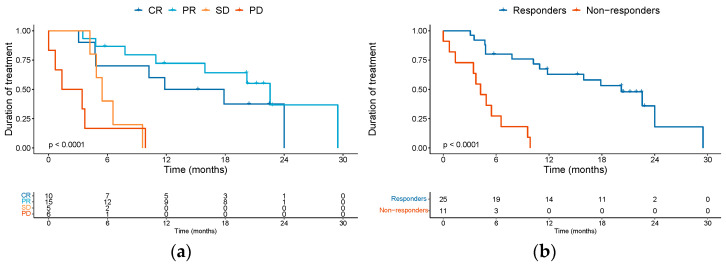
Kaplan–Meier curves depicting median treatment duration in months according to BOR (**a**) and comparing responders (CR + PR) versus non-responders (SD + PD) (**b**). Abbreviations: BOR: best overall response; CR: complete response; PR: partial response; SD: stable disease; PD: progression of disease.

**Figure 2 cancers-15-04312-f002:**
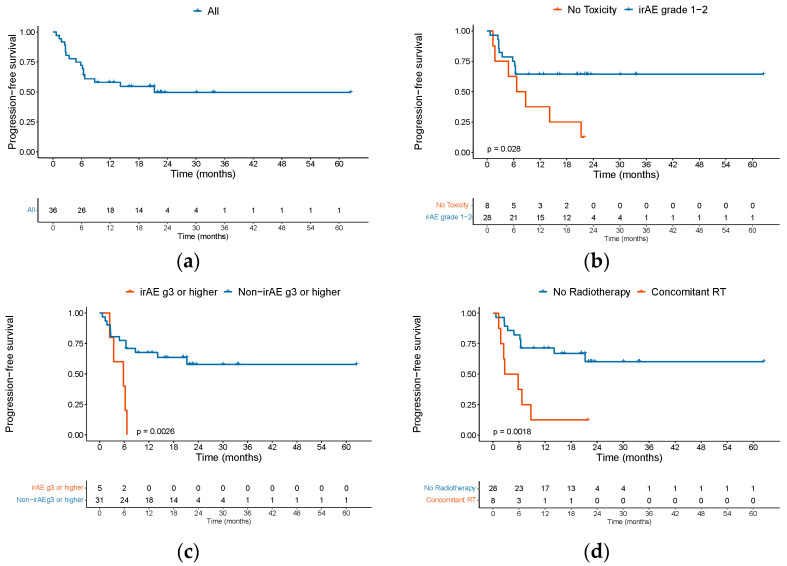
PFS among patients with advanced cSCC treated with ICI. Kaplan–Meier curves showing (**a**) PFS of the entire cohort in months, (**b**) PFS between patients presenting grade 1–2 irAE versus no toxicity, (**c**) PFS between patients presenting grade 3 or higher irAE versus non-irAE grade 3 or higher, and (**d**) PFS for patients that received concomitant RT versus none. Abbreviations: PFS: progression-free survival; cSCC: cutaneous squamous cell carcinoma; ICI: immune checkpoint inhibitor; irAE: immune-related adverse event; RT: radiation therapy.

**Figure 3 cancers-15-04312-f003:**
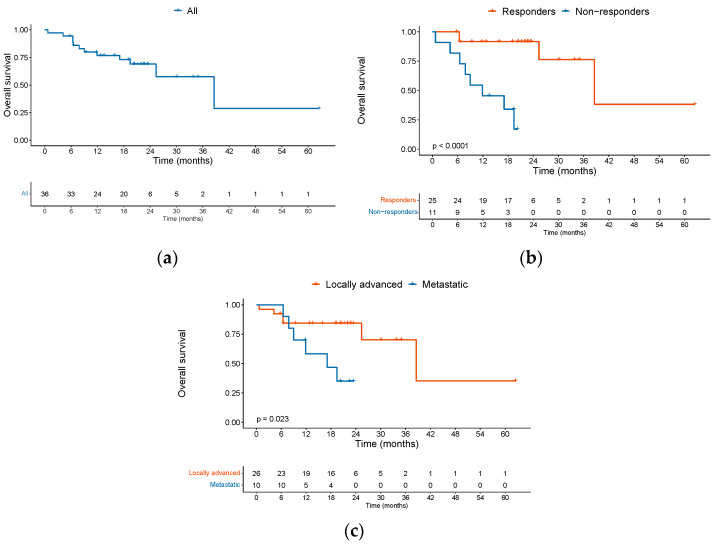
OS among patients with advanced cSCC treated with ICI. Kaplan–Meier curves showing (**a**) OS of the entire cohort in months, (**b**) OS according to BOR responders (CR + PR) and non-responders (SD + PD), and (**c**) OS according to extent of disease locally advanced versus distant metastasis. Abbreviations: OS: overall survival; cSCC: cutaneous squamous cell carcinoma; ICI: immune checkpoint inhibitor; BOR: best overall response; CR: complete response; PR: partial response; SD: stable disease; PD: progression of disease.

**Table 1 cancers-15-04312-t001:** Baseline characteristics of our population.

Characteristics—*n* (%)	SCC (*n* = 36)
Sex	
Female	9 (25)
Male	27 (75)
Age	
Median (min-max)/IQR	75.4 (27.9 to 100.1)/(72.4 to 84.4)
27.9 to 69 years of age	6 (16.7)
70 to 79 years of age	17 (47.2)
80 to 89 years of age	8 (22.2)
90 to 100.1 years of age	5 (13.9)
ECOG performance status	
0	10 (27.8)
1	16 (44.4)
2	8 (22.2)
3	2 (5.6)
Comorbidity	
Rheumatological disease on IS drug	2 (5.6)
Solid organ transplant recipient	2 (5.6)
Hematological malignancy	10 (27.8)
EB	2 (5.6)
None	20 (55.6)
Primary site	
Head and neck	25 (69.4)
Limbs	6 (16.7)
Torso	2 (5.6)
Unknown	3 (8.3)
Primary treatment	
Surgery	21 (58.3)
Surgery + adjuvant RT	9 (25)
RT alone	3 (8.3)
ICI	2 (5.5)
Other systemic therapy	1 (2.8)
Extent of disease	
Locally advanced/Unresectable	26 (72.2)
Distant metastasis	10 (27.8)
AJCC clinical stage at ICI start	
Recurrent Stage I	1 (2.8)
Recurrent Stage II	3 (8.3)
Recurrent Stage III	7 (19.4)
Stage IV at presentation	4 (11.1)
Recurrent stage IV	21 (58.3)
ICI line of therapy	
First-line	35 (97.2)
Second line	1 (2.8)
Concomitant radiation therapy	
No	28 (77.8)
Concurrent to ICI at ICI start	3 (8.3)
Completed in the 2 weeks pre-start of ICI	1 (2.8)
Concurrent to ICI for oligoprogression of disease	4 (11.1)

Abbreviations: IQR: Interquartile range; ECOG: Eastern Cooperative Oncology Group; IS: immunosuppressant; EB: epidermolysis bullosa; RT: radiotherapy; ICI: immune checkpoint inhibitor; AJCC: American Joint Committee on Cancer 8th edition.

**Table 2 cancers-15-04312-t002:** Reason for treatment discontinuation (N = 26).

Disease progression	8 (30.8%)
Adverse reactions	5 (19.2%)
Achieved maximum benefit	5 (19.2%)
Maximum number of doses	3 (11.5%)
Other	3 (11.5%)
Death	2 (7.7%)

**Table 3 cancers-15-04312-t003:** Univariable Analysis for PFS.

Variable		HR	95% CI	*p* Value
Sex	Female (ref.)			
	Male	0.796	0.28–2.65	0.669
Age	<75 years (ref.)			
	≥75 years	1.073	0.39–2.91	0.890
ECOG	0–1 (ref.)			
	≥2	1.043	0.37–2.96	0.937
Grade 1–2 toxicity	No (ref.)			
	Yes	0.357	0.14–0.94	0.037
Grade ≥ 3 toxicity	Yes (ref.)			
	No	0.210	0.06–0.65	0.006
Scenario	Localized (ref.)			
	Metastatic	1.645	0.61–4.46	0.328
Comorbidities	No (ref.)			
	Yes	0.930	0.36–2.41	0.881
BOR	Responders (ref.)	2.996		
	Non-responders		1.82–4.93	0.000002

**Table 4 cancers-15-04312-t004:** Univariable analysis for OS.

Variable		HR	95% CI	*p* Value
Sex	Female (ref.)			
	Male	0.835	0.933	0.35–4.95
Age	<75 years (ref.)			
	≥75 years	0.22–3.15	1.185	0.695
ECOG	0–1 (ref.)			
	≥2	0.790	0.35–3.98	0.291
Grade 1–2 toxicity	Yes (ref.)			
	No	1.054	0.784	0.07–1.18
Grade ≥ 3 toxicity	Yes (ref.)			
	No	0.31–3.62	1.306	0.085
Scenario	Localized (ref.)			
	Metastatic	3.896	1.10–13.87	0.036
Comorbidities	No (ref.)			
	Yes	0.487	0.15–1.63	0.244
BOR	Responders (ref.)			
	Non-responders	12.85	2.67–61.84	0.001

## Data Availability

The data presented in this study are available on request from the corresponding author. The data are not publicly available due to protection of patient information.

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
