# Peer review of "Immune Checkpoint Inhibitors in Advanced Cutaneous Squamous Cell Carcinoma: Real-World Experience from a Canadian Comprehensive Cancer Centre"

_cancers, 2023, doi:10.3390/cancers15174312_

Round 1
Reviewer 1 Report
The manuscript presents valuable data on the effectiveness and safety of anti-PD-1 therapy in locally advanced or metastatic cSCC. However, addressing the mentioned points would significantly improve the manuscript's quality and impact. I recommend revision and re-submission after careful consideration of the suggested improvements. all of the points below have been mentioned in my comments to the authors
Comments for authors:
Title:
The title accurately reflects the content of the study and is concise and informative.
Abstract:
1. While the abstract reports the overall response rates (41.7% partial response, 27.8% complete response, 13.9% stable disease), it would be helpful to include confidence intervals or standard errors to convey the precision of these estimates.
2. Acknowledge the limitations of the study, such as the small sample size and retrospective design, in the abstract to ensure the readers are aware of potential biases and constraints.
Introduction:
1. The introduction provides a good overview of cutaneous squamous cell carcinoma (cSCC) and its prevalence but lacks specific information on the research gap or objective of the study. It would be helpful to clearly state the research question or hypothesis being addressed.
2. In the introduction, there is a mention of a recent study that cited a 1:1 ratio between basal cell carcinoma and cSCC [2]. However, it would be beneficial to provide more details about this study, such as its methodology and sample size, to establish the credibility of the citation.
3. The introduction highlights the recent FDA approval of cemiplimab and pembrolizumab for advanced cSCC [58-59], but it does not elaborate on the safety and efficacy data from the pivotal trials that led to these approvals. Including a brief summary of the key findings from these trials would strengthen the introduction.
4. Consider adding a sentence or two at the end of the introduction to outline the specific aims and contributions of the current study, emphasizing the need to investigate the efficacy and safety of anti-PD-1 ICI in the Canadian real-world cohort.
Methods:
1. The section does not mention the total number of patients included in the cohort, which is crucial information for understanding the size and scope of the study.
2. The criteria used to determine the best overall response (BOR) by the investigator should be explicitly stated or referenced to ensure transparency and reproducibility.
3. Elaborate on the electronic pharmacy record query process used to identify the patient cohort, including details on the search terms and filters applied, to ensure the accuracy and comprehensiveness of patient selection.
4. Expand on the comorbidities of interest (e.g., immunosuppressive conditions, genetic syndromes predisposing to cSCC) and their relevance to the study's research question. Provide references or existing literature supporting their inclusion in the analysis.
5. Provide a clear and detailed rationale for grouping patients with disease compromising the immune system and patients on immunosuppression together as "immunocompromised." Include references that support this classification and explain its importance for answering the research question.
Results:
1. Lack of Comparison Group: The absence of a control or comparison group limits the ability to draw definitive conclusions about the efficacy and safety of the treatments. A comparison group of patients receiving standard care or alternative treatments would allow for better interpretation of the results
2. Data Collection: It is not entirely clear how the data were collected and if any potential biases were considered during the retrospective analysis. More information on data collection methods, blinding, and validation processes would enhance the study's rigor.
3. The safety results seem somewhat superficial. It would be helpful to include more comprehensive information about the incidence of adverse events, their management, and their impact on treatment continuation and patient outcomes.
4. The Kaplan-Meier curves are informative, but a more in-depth analysis of survival outcomes, such as Cox proportional hazards models, adjusting for potential confounding factors, would strengthen the study's conclusions.
Discussion:
1. The opening statement about the importance of multi-disciplinary management for advanced cSCC is reasonable, but it lacks specific references to relevant literature or studies supporting this claim. Incorporating evidence from relevant studies that demonstrate the benefits of multi-disciplinary approaches would enhance the credibility of this statement.
2. The statement that systemic treatment with anti-PD-1 is the new standard of care for advanced cSCC patients not eligible for curative intent surgery or radiation therapy needs further support. The reference to phase I and II clinical trials is given, but it would be helpful to provide more recent and comprehensive phase III trial data and systematic reviews/meta-analyses confirming this conclusion.
3. The authors compare their study results to previous phase I and II trials, but they only mention a range of ORRs reported in those trials. To make this comparison more robust, the authors should provide specific references to individual phase I and II trials with corresponding ORRs and CR rates.
4. When discussing the impact of immune status (compromised versus competent) on response rate and survival outcomes, the authors state that their study and other real-world studies found no significant difference. However, they should provide references to those real-world studies to strengthen this claim.
5. The discussion suggests that combining radiation therapy with immunotherapy could act as an "accelerant," but the statement lacks specific references to relevant studies supporting this claim. Citing studies that explore the potential synergistic effects of radiation therapy and immunotherapy in cSCC patients would be beneficial.
Conclusions:
1. It would be helpful to provide some context on the current treatment landscape for advanced cSCC. Compare the results of the study with existing treatment options to emphasize the contribution of anti-PD-1 therapy.
2. The conclusion states that the study supports the use of anti-PD-1 as a first-line treatment, but it doesn't explain the rationale behind this recommendation. Including a brief summary of the advantages of anti-PD-1 therapy over other treatments and its potential impact on patient outcomes would strengthen the conclusion.
3. Consider mentioning any potential areas for future research or investigation based on the findings of this study. This can help guide other researchers and provide a sense of direction for the field.
Reviewer 2 Report
Very interesting study. The two main limitations are the limited sample size and the retrospective design. The authors should comment that the sutdy could be underpowered to detect the outcomes.
Some figures showing the response of the lesions after treatment could be useful.
The authors should comment more on the safety profile of these drugs, with particular reference to the risk of flare of pre-existing immune diseases (cite the recent MA: PMID: 33314269 )
Round 2
Reviewer 1 Report
No further comments! Thank you for your hard work in addressing my comments.
Reviewer 2 Report
The revised version of the manuscript is OK. Thank you!